# Effects of Polyunsaturated Fatty Acid/Saturated Fatty Acid Ratio and Different Amounts of Monounsaturated Fatty Acids on Adipogenesis in 3T3-L1 Cells

**DOI:** 10.3390/biomedicines12091980

**Published:** 2024-09-02

**Authors:** Sim Yee Lim, Yi-Wen Chien

**Affiliations:** 1School of Nutrition and Health Sciences, Taipei Medical University, Taipei 11031, Taiwan; simyee03117@gmail.com; 2Graduate Institute of Metabolism and Obesity Sciences, Taipei Medical University, Taipei 11031, Taiwan; 3Nutrition Research Center, Taipei Medical University Hospital, Taipei 11031, Taiwan; 4Research Center of Geriatric Nutrition, College of Nutrition, Taipei Medical University, Taipei 11031, Taiwan

**Keywords:** polyunsaturated to saturated fatty acid ratio, monounsaturated fatty acids, adipocytes, adipogenesis

## Abstract

(1) Background: Adipose tissue serves as a central repository for energy storage and is an endocrine organ capable of secreting various adipokines, including leptin and adiponectin. These adipokines exert profound influences on diverse physiological processes such as insulin sensitivity, appetite regulation, lipid metabolism, energy homeostasis, and body weight. Given the integral role of adipose tissue in metabolic regulation, it is imperative to investigate the effects of varying proportions and types of dietary fats on adipocyte function. In addition, our previous study showed that P/S = 5 and MUFA = 60% appeared to be beneficial in preventing white adipose tissue accumulation by decreasing plasma insulin levels and increasing hepatic lipolytic enzyme activities involved in β-oxidation. Therefore, the objective of this study was to explore the effects of a polyunsaturated fatty acid (PUFA) to saturated fatty acid (SFA) ratio of 5 and varying levels of monounsaturated fatty acids (MUFA = 30% or 60%) on lipogenesis. (2) Methods: We cultured 3T3-L1 mouse embryo fibroblasts in Dulbecco’s modified Eagle’s medium (DMEM) containing 10% bovine calf serum until confluent. Varying ratios of palmitic acid (PA), oleic acid (OA), and linoleic acid (LA) were first bound with bovine serum albumin (BSA) before being applied to 3T3-L1 adipocytes in low doses and in high doses. (3) Results: Low doses of P/S ratio = 5, MUFA = 60% (M60) fatty acids decreased the accumulation of triglycerides in mature adipocytes by decreasing the mRNA expression of adipogenic factors, such as peroxisome proliferator-activated receptors (PPARs), lipoprotein lipase (LPL), and glucose transporter-4 (GLUT-4), while increasing lipolytic enzyme (hormone-sensitive lipase, HSL) expression when compared to high doses of P/S ratio = 5, MUFA = 60% (M60), low and high doses of P/S ratio = 5, MUFA = 30% (M30). Furthermore, the treatment of M60 in low doses also decreased the secretion of leptin and increased the secretion of adiponectin in adipocytes. (4) Conclusions: The composition of P/S = 5, MUFA = 60% fatty acid in low doses appeared to result in anti-adipogenic effects on 3T3-L1 adipocytes due to the down-regulation of adipogenic effects and the transcription factor.

## 1. Introduction

The scientific community recognizes obesity as the most important health problem related to the genesis of metabolic syndrome [1]. The increase in the frequency of obesity and its comorbidities has been observed worldwide, not only in developed countries such as the United States of America (U.S.A.) but also in Taiwan. In fact, the World Health Organization (WHO) estimates that there are 890 million obese and 2.5 billion overweight individuals globally [2]. These estimates highlight the critical importance of identifying a solution to this major worldwide problem.

Maintaining adequate body weight requires balancing food consumption with energy expenditure. Failing to maintain this balance results in obesity. Dietary interventions may induce changes in this metabolic and inflammatory state by modulating the expression of important genes involved in these chronic manifestations. We believe that regulation of gene expression by dietary fats most severely impacts the development of obesity and insulin resistance. Therefore, as dietary fat contains more calories than protein or carbohydrates, we recommend limiting the intake of fat in order to prevent obesity.

Several official institutions, including the WHO and the American Heart Association (AHA), recommend low-fat diets as a prophylactic treatment for various lipid-related disorders. However, reducing the relative amount of fat in the diet alone appears to be insufficient to prevent weight gain. In the U.S.A., energy intake from dietary fats dropped from 40% to 33% from the 1960s to 1995. In the same period, the number of obese adults (BMI > 25) steadily increased, reaching 56% in the early 1990s and 65% at present [3]. Similarly, in Norway, the relative intake of dietary fats dropped from 40% to 34% from the middle of the 1970s to the middle of the 1990s. In spite of this decrease in intake of dietary fats, the average male BMI increased from 24.8 to 26.6 (9.1 kg), and the average female BMI increased from 24.7 to 25.1 (3.7 kg) from the 1960s to 1999. Therefore, we suggest that not only the quantity of ingested fats but also the composition of fatty acids maintain importance for human health.

The amount and type of dietary fatty acids can regulate complex intracellular signaling systems, thereby modulating cellular metabolism. Remarkably, dietary fat contributes to both the phospholipid composition of the adipocyte membranes and the modulation of the transcription of different genes involved in the processes of lipolysis and lipogenesis. Moreover, fatty acids significantly influence adipocyte regulation as they affect the secretion of various adipokines and inflammatory biomarkers.

Adipocytes play a central role in maintaining lipid homeostasis and energy balance by storing triglycerides or releasing free fatty acids in response to changes in energy demands. Because the growth of adipose tissue can result from both hyperplasia and hypertrophy of adipocytes [4], several studies have focused on the process of adipocyte proliferation and differentiation [5,6].

Research has predicted that fatty acids influence PPARγ-regulated gene expression and subsequent lipid droplet formation during adipocyte differentiation [7,8]. Recently, researchers have noted an influence on adipocyte size by the fatty acid composition in the lipid droplets. This observed influence is regulated by the delta-9 desaturase/stearoyl Co-A desaturase 1 (*D9D/SCD1*) gene, which is PPARγ-dependent [9]. Thus, some bioactive lipids, such as medium-chain fatty acids, conjugated fatty acids, and n-3 and n-6 polyunsaturated fatty acids (PUFAs), are involved in lipid storage and the physiologic functions of lipid metabolism [10], whereas cooking oil contains only trace amounts of these bioactive lipids. However, diets containing a high polyunsaturated-to-saturated fatty acid (P/S) ratio could increase postprandial fat oxidation [11]. Therefore, defining the roles of the P/S ratio and monounsaturated fatty acids (MUFAs) on triglyceride accumulation in 3T3-L1 adipocytes is vital.

Several studies have indicated that the influences of MUFAs on regulating body weight are controversial between human trials and animal models. For example, one study indicated that rats fed with olive oil diets maintained a higher rate of body weight gain and abdominal fat deposition than those fed with safflower or lard oil diets [12]. Furthermore, olive oil, which consists of significant amounts of MUFAs, has demonstrably increased hepatic lipogenic enzyme activity in animal studies, whereas sunflower and linseed oils, which contain a large amount of PUFAs, have decreased lipogenic enzyme activities [13]. However, Clifton et al. [14] reported that the effects of high-fat diets rich in MUFAs (canola and high-oleic sunflower oils) on weight loss resemble the effects of low-fat diets; however, it would remain the secretion of leptin in a human trial.

In addition, our previous study showed that P/S = 5 and MUFA = 60% appeared to be beneficial in preventing white adipose tissue accumulation by decreasing plasma insulin levels and increasing hepatic lipolytic enzyme activities involved in β-oxidation. Therefore, determining whether the P/S ratio and MUFA percentage relate to triglyceride accumulation in 3T3-L1 adipocytes is essential. In this study, we compare the effects of the P/S ratio (P/S ratio = 5) and two different MUFA percentages (60% and 30%) on the mechanism of fat accumulation by measuring gene expressions related to lipid metabolisms, such as PPARγ, lipoprotein lipase (LPL), hormone-sensitive lipase (HSL), adiponectin, leptin, and glucose transporter-4 (GLUT-4).

## 2. Materials and Methods

### 2.1. Materials and Cell Culture

3T3-L1 mouse embryo fibroblasts were cultured in Dulbecco’s modified Eagle’s medium (DMEM) containing 10% bovine calf serum until confluent. Two days after post-confluence, we stimulated the cells to differentiate, utilizing 0.5 mmol/L isobutylmethylxanthine, 1 mmol/L dexamethasone, and 10 mg/mL insulin added to DMEM containing 10% fetal bovine serum (FBS) for 6 d. We then maintained the cells in 10% FBS/DMEM medium with 10 mg/mL insulin for another 6 d, followed by culture with 10% FBS/DMEM medium until analysis. Approximately 13 d after induction of differentiation, 90% of the cells displayed the characteristic lipid-filled adipocyte phenotype. All of the fatty acids were delivered to the cells as fatty acid/bovine serum albumin (BSA) complexes. All mediums contained 100 kU/L penicillin and 100 mg/L streptomycin, which we maintained at 37 °C in a humidified 5% CO_2_ atmosphere.

The P/S ratio = 5 and MUFA = 30% or 60% were achieved using the method below. The dosage of 10–200 µM refers to the concentration of fatty acid in the combination of the two groups below:(a)P/S ratio = 5; MUFA 60% was prepared by using the volume ratio of palmitic acid: oleic acid: linoleic acid equals 1:10:5;(b)P/S ratio = 5; MUFA 30% was prepared by using the volume ratio of palmitic acid: oleic acid: linoleic acid equals 1:2.5:5.

### 2.2. Cell Viability Assay

We performed an MTT assay to determine the number of viable cells in the culture. Then, we added different concentrations (0, 10, 30, 50, 100, and 200 µM) of fatty acids along with the differentiation medium to the adipocytes. Afterward, we incubated the cells for 72 h. At the appointed time, we measured the absorbance at 570 nm in a plate reader to determine the formazan concentration, which is proportional to the number of living cells in a culture.

### 2.3. Cellular Fatty Acid Composition

We extracted the total lipids from the cell lysates and modified them according to Folch et al. [15], after which we prepared the fatty acid methyl esters for analysis by gas–liquid chromatography. Fatty acid analysis was performed using Trace Gas Chromatography (GC) produced by ThermoQuest Company, and the analysis results were integrated using Trade ChromCard software version 1. Oven: The starting temperature is 160 °C, rising at a rate of 1 °C per minute to 250 °C. Syringe: The injection temperature is 260 °C, the split speed is 20 mL per minute, and the split ratio is 20. Carrying gas: The mobile phase is nitrogen, and the flow rate is 2 mL per minute. Detector: The operating temperature of the flame ionization detector (FID) is 260 °C, the airflow rate is 350 mL per minute, the hydrogen flow rate is 40 mL per minute, and the nitrogen flow rate is 30 mL per minute. At this point, we identified the fatty acids by comparing the retention times of the samples with those of a known standard. Fatty acid composition values are expressed as a percentage of total fatty acids.

### 2.4. Oil Red O Staining

After 72 h of treatment, we washed the adipocytes with cold, phosphate-buffered saline (PBS; pH 7.4) and fixed them with a 4% paraformaldehyde solution in PBS. Following this procedure, we stained the cells with freshly prepared Oil Red O dye (0.5% (*w*/*v*)) dissolved in isopropanol and diluted at a 3:2 ratio of dye to water. Prior to quantification, we washed the cells thoroughly with distilled water. We then performed a spectrophotometric analysis of the stain by dissolving the stained lipid droplets with isopropanol and measuring at 492 nm. We calculated the values as a percentage of the vehicle control and expressed them as means ± SEM (*n* = 3).

### 2.5. Lipolysis Assay

We treated the fully differentiated adipocytes for 72 h with 10 µM or 100 µM of the fatty acids grouped as M60 and M30, respectively. Afterward, we removed the conditioned medium from each well and assayed them for glycerol content with a free glycerol determination kit (RANDOX GY105, Crumlin, UK).

### 2.6. RNA Isolation and Gene Expression Quantification

We extracted the total RNA from the 3T3-L1 adipocytes utilizing Trizol. Subsequently, we measured the purity of the RNA at A260 nm/A280 with ratio values of 1.8~2.0. We then synthesized the cDNA from 1 μg of RNA utilizing a high-capacity RNA to cDNA reverse transcriptase kit in a total of 12 μL of reaction volume. Afterward, we performed reverse transcription with sample incubation at 42 °C for 1 h, followed by 70 °C for 5 min and 4 °C for another 5 min. All PCR reactions were performed utilizing a thermal cycler (7300 Applied Biosystems, Waltham, MA, USA). To measure relative quantification, we employed the ΔΔCT method, normalizing values to the reference gene (β-actin). Primers used in this study included adiponectin (forward (F): 5′-TGTAGGATTGTCAGTGGATCTG-3′ and reverse (R): 5′-GCTCTTCAGTTGTAGTAACGTCATC-3′), GLUT4 (F: 5′-ACTCTTGCCACACAGGCTCT-3′ and R: 5′-AATGGAGACTGATGCGCTCT-3′), HSL (F: 5′-AGGCCTTGTGTTGTGTTTCCA-3′ and R: 5′-TGGGGGACAGCTTCCTTTCTT-3′), LPL (F: 5′-ATCCATGGATGGACGGTAACG-3′ and R: 5′-CTGGATTCCAATACTTCGACCA-3′), PPARγ (F: 5′-TTTTCAAGGGTGCCAGTTTC-3′ and R: 5′-AATCCTTGGCCCTCTGAGAT-3′), leptin (F: 5′-TGGACCAGACTCTGGCAGTC-3′ and R: 5′-AGGACACCATCCAGGCTCTC-3′) and β-ACTIN (F: 5′-ACAGGATGCAGAAGGAGATTAC-3′ and R: 5′-ACAGTGAGGCCAGGATAGA-3′).

### 2.7. Apoptosis Assay

For the detection of apoptosis in 3T3-L1 cells, we employed a single-stranded DNA ELISA kit (ApoStrand, BIOMOL, Plymouth Meeting, PA, USA). This assay is based on the selective denaturation of DNA in apoptotic cells by formamide, which reflects changes in chromatin associated with apoptosis. We added varying concentrations of M30 and M60 to the adipocytes with the medium. After 72 h of culture, the cells were fixed, followed by the procedure described by the kit. At that point, we utilized an ELISA plate reader to measure the absorbance at 405 nm.

### 2.8. Statistical Analysis

We expressed the results as means ± SEM. Employing a one-way analysis of variance (ANOVA) and a *t*-test, utilizing the SPSS package program version 18.0, we determined the statistical significance of differences among the groups. We considered the results significant if the value of *p* was <0.05 using the Duncan post hoc test.

## 3. Results

### 3.1. Effect of Different Concentrations of M60 and M30 Fatty Acid Composition on Adipocyte Viability

The five concentrations (10, 30, 50, 100, and 200 µM) of M60 and M30 treatments on mature adipocytes did not affect cell viability within 24 h (Figure 1A). A total of 50 µM and 200 µM of the M60 treatment could cause the cell viability to significantly decrease compared with the control group in 48 h (Figure 1B). Therefore, we chose 10 µM as the low-dose group and 100 µM as the high-dose group since these doses would not cause any changes in cell viability when compared with the control.

### 3.2. Effect of Different Concentrations of M60 and M30 Fatty Acid Compositions on Triglyceride Accumulation

The representative images of Oil Red O staining demonstrated that low doses of the M60 treatment (M60-10) could suppress lipid accumulation compared with the control and other groups (Figure 2A). Thus, we observed the same phenomenon in triglyceride content. The triglyceride content of the M60-10 group significantly decreased when compared with the control group (Figure 2B).

### 3.3. Effect of Different Concentrations of M60 and M30 Fatty Acid Compositions on Adipocyte Lipolysis

Low doses of the M60 and M30 groups increased basal lipolysis in fully differentiated 3T3-L1 adipocytes compared with the high-dose group (Figure 3). Thus, the glycerol released in both high-dose groups (M60-100 and M30-100) significantly decreased compared with the control after 72 h of incubation.

### 3.4. Gene Expression of Adipogenic Enzymes

Figure 4 illustrates the impact of different concentrations of the M60 and M30 fatty acid compositions on PPARγ, LPL, and GLUT-4 mRNA expression. Adding the M60-10 treatment to mature adipocytes significantly decreased the PPARγ mRNA expression by nearly 50% compared with the M60-100, M30-10, and M30-100 groups; however, no significant differences existed as compared to the control group (Figure 4A). While the LPL and GLUT-4 gene expressions maintained similar results to those of the PPARγ mRNA, the M60-10 treatment significantly decreased the LPL and GLUT-4 mRNA expression compared with the M60-100, M30-10, and M30-100 groups (Figure 4B,C). Furthermore, the LPL mRNA expression of M60-10 was significantly lower than that of the control group.

### 3.5. Gene Expression of Lipolytic Enzymes

As shown in Figure 5, there was a significantly increased expression of HSL genes in mature adipocytes treated with M60-10 compared with the control, M60-100, M30-10, and M30-100 groups (*p* < 0.05). Thus, the HSL mRNA expression of both the M60-100 and M30-10 groups was also significantly higher than that of the control group (Figure 5).

### 3.6. Gene Expression of Adipokines Secreted by the Adipocytes

Figure 6A illustrates that the leptin mRNA expression of M60-10 significantly decreased compared with the control, M60-100, M30-10, and M30-100 groups. On the other hand, there was a significantly increased expression of adiponectin genes in mature adipocytes treated with M60-10 compared with the control and M30-10 groups (Figure 6B). Thus, the adiponectin gene expression in both high-dose groups (M60-100 and M30-100) was also significantly higher than that in the control group.

### 3.7. Fatty Acid Uptake of Every Single Fatty Acid

To investigate the intracellular accumulation of fatty acids, gas chromatography was employed to analyze the fatty acid composition of adipocytes following 72 h of fatty acid treatment. Figure 7 shows that the cellular fatty acid content of palmitic acid, oleic acid, and linoleic acid in mature adipocytes increased slightly after 72 h of treatment as compared with 0 h, indicating cellular uptake of the administered fatty acids over the 72 h period.

### 3.8. Effect of Different Ratios and Concentrations of Fatty Acids on Apoptosis

We examined whether the inhibitory effect of the M60-10 group on triglyceride accumulation could be related to apoptosis. After 72 h of treatment, M60-10 clearly increased apoptosis in the mature adipocytes (Figure 8). This result indicates that the decrease in triglyceride content is due, at least in part, to the effect of the M60-10 group on inducing apoptosis in adipocytes.

## 4. Discussion

We found that oil with a P/S ratio equal to 5 and a MUFA percentage of 60% is important for decreasing triglyceride content in mature 3T3-L1 adipocytes. A previous study indicated that a high P/S ratio could increase postprandial lipid oxidation, basal metabolic rate, and thermogenesis [16]. Furthermore, in a high-fat diet, high MUFA increases lipid oxidation and energy expenditure [17,18]. We speculated that the reason M60-10 could decrease triglyceride accumulation when compared with other groups was its high MUFA content.

Triglyceride accumulation was affected by three pathways, which included the transcription factor PPARγ expression, the fatty acid synthesis pathway, and the lipolysis pathway. When the expression of PPARγ increases, triglyceride accumulation also increases. Similarly, when gene expression related to fatty acid pathways, such as LPL and GLUT-4, increases, the triglyceride content in adipocytes increases as well. On the other hand, when the HSL mRNA expression related to the lipolysis pathway increases, lipolysis occurs, decreasing triglyceride accumulation.5 PPARγ is always found in adipocytes. When PPARγ is activated, differentiation of pre-adipocytes into adipocytes occurs, and triglycerides accumulate in the cell [19]. In this study, the PPARγ gene expression in the M60-10 group was significantly lower than that in the M30-10 group. This result was consistent with a previous study that concluded that PPARγ not only plays the role of the transcription factor but also the activator of fatty acid synthesis [20,21,22,23]. The researchers found GLUT-4 in insulin-sensitive tissue, i.e., muscle tissue and adipose tissue. When GLUT-4 is stimulated, it increases glucose uptake in specific tissues [24]. A previous study by Field indicates that increasing intrinsic GLUT-4 results in glucose uptake in 3T3-L1 adipocytes [25]. Field also states that PUFA could increase lipid oxidation, thermogenesis, insulin sensitivity, and glucose uptake in adipose tissue [26]. As a result, we speculate that the fatty acid composition of M60-10, which is high in P/S ratio and MUFA, could decrease PPARγ gene expression and, thus, decrease the activation of GLUT-4 and LPL mRNA expression when compared with M60-100, M30-10, and M30-100.

Modifications in lipolysis pathways may influence lipid droplet formation and triglyceride accumulation. HSL represents the major rate-limiting lipase for lipolysis in adipocytes. A previous study indicates that lipolysis in a high-fat diet is lower than that in a low-fat diet, which is similar to the result that we obtained. Furthermore, the author states that the mRNA and protein expression of HSL are not influenced by the composition of fatty acids but rather by the total quantity of fats [27]. Therefore, increased lipid mobilization from adipocytes by lipolysis could constitute another explanation for the reduction in triglyceride content in 3T3-L1 adipocytes. To more accurately assess lipolysis, it is recommended to further examine the protein levels of HSL, ATGL, and MGL in future studies.

Adiponectin, an anti-inflammatory and insulin-sensitizing protein secreted by adipose tissue, may be modulated by dietary fatty acids. The level of adiponectin is inversely correlated with obesity and obesity-associated complications. Impaired adiponectin signaling, caused by decreased expression of adiponectin or adiponectin receptors, leads to insulin resistance [28,29]. Our finding indicates that a low dose of the M60 fatty acid composition upregulated adiponectin in 3T3-L1 adipocytes, results that are consistent with those of Yadav et al., who observed that a high-fat diet could cause a decrease in adiponectin [30]. On the other hand, a previous study observed a positive correlation between adipose depots and leptin levels. Leptin levels in the white adipose tissue and plasma are related to the energy stores, such that leptin increases during a state of obesity and decreases during fasting. Leptin causes stimulation of appetite and inhibition of sympathetic nerve activity [31,32]. Circulating leptin crosses the blood–brain barrier and mediates its action through the Janus kinase (JAK) signal transducer and activator of the transcription (STAT) pathway (JAK–STAT3) [33]. Our study is consistent with the Havel et al. study [34], which found that leptin secretion is correlated with glucose utilization in adipocytes. We find that the GLUT-4 gene expression in M60-10 is lower than that in the M30-10 group, which is similar to leptin gene expression.

In addition, apoptosis in adipocytes could contribute to the reduction in cellularity. The decrease in triglyceride accumulation may be accompanied by the initiation of apoptosis [35]. Our single-stranded DNA ELISA results demonstrate a pronounced apoptotic effect of a low dose of the M60 fatty acid composition (M60-10) after 72 h of treatment. Considering that 3T3-L1 adipocytes undergo induction of apoptosis in cells, there may be a mechanism by which the M60-10 group attenuates adipogenesis, leading to reduced lipid accumulation.

## 5. Conclusions

In conclusion, these data demonstrate that a low dose of an M60 fatty acid composition (P/S ratio = 5, MUFA 60%) acts primarily by reducing the gene expression related to adipogenesis, such as PPARγ, LPL, GLUT-4, and leptin, and by increasing gene expression of HSL and adiponectin, thus inducing apoptosis. Hence, such a dose could mediate a reduction in lipids.

## Figures and Tables

**Figure 1 biomedicines-12-01980-f001:**
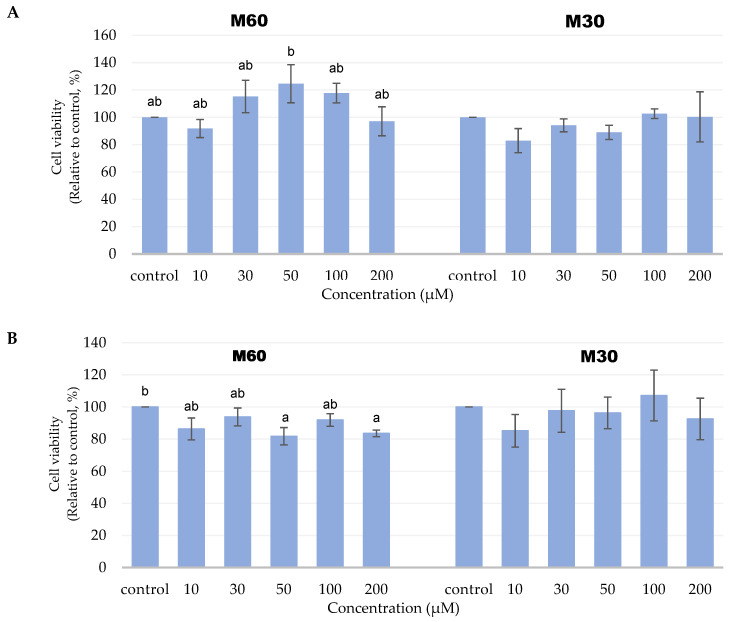
Cell viability of 3T3-L1 adipocytes after adding different ratios and concentrations of mixed oil for (**A**) 24 and (**B**) 48 h, respectively, by using the MTT assay. Values are expressed as the mean ± SEM (*n* = 3). Values with different letters are significantly different at *p* < 0.05, as measured by a one-way ANOVA followed by a Duncan post hoc test.

**Figure 2 biomedicines-12-01980-f002:**
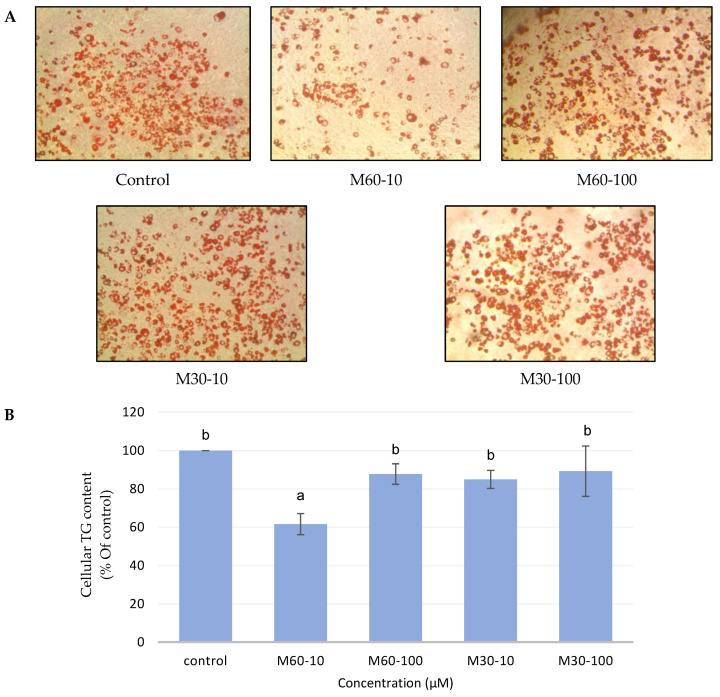
Effects of adding different ratios and concentrations of mixed fatty acids for 72 h on adipocyte morphology and triglyceride content in 3T3-L1 adipocytes. (**A**) The cell morphology of differentiated adipocytes was analyzed with phase contrast microscopy after Oil Red O staining. (**B**) The triglyceride content of 3T3-L1 adipocytes after adding different mixed fatty acids for 72 h. Values are expressed as the mean ± SEM (*n* = 3). Values with different letters are significantly different at *p* < 0.05, as measured by a one-way ANOVA followed by a Duncan post hoc test.

**Figure 3 biomedicines-12-01980-f003:**
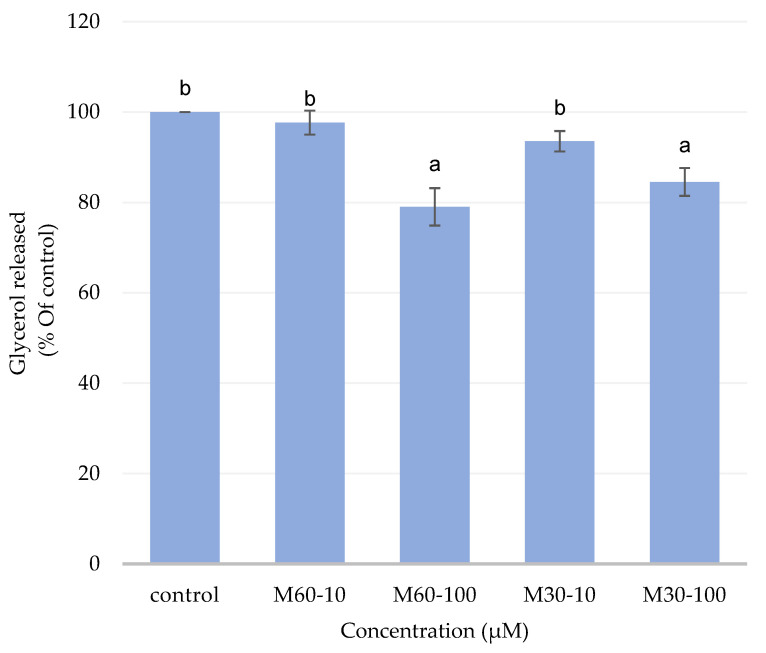
Effects of adding different ratios and concentrations of mixed fatty acids for 72 h on glycerol content in 3T3-L1 adipocytes. Values are expressed as the means ± SEM (*n* = 3). Values with different letters are significantly different at *p* < 0.05, as measured by a one-way ANOVA followed by a Duncan post hoc test.

**Figure 4 biomedicines-12-01980-f004:**
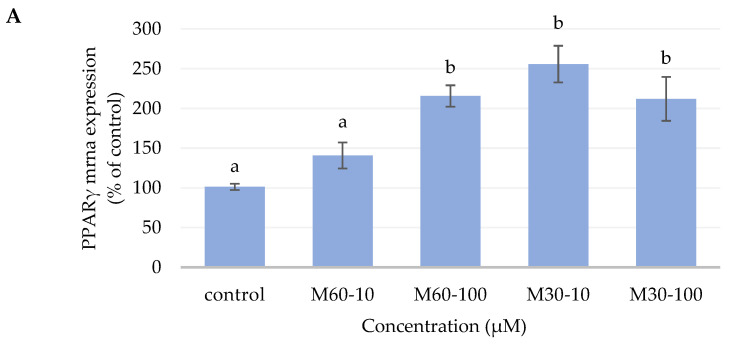
Semi-quantitative RT-PCR analysis for adipogenic enzymes (**A**) PPARγ, (**B**) LPL, and (**C**) GLUT-4 mRNA levels in 3T3-L1 adipocytes after adding different ratios and concentrations of mixed oil for 72 h. The graphs represent the mRNA levels for PPARγ, LPL, and GLUT-4 relative to the control gene, β-actin. Values are expressed as the mean ± SEM (*n* = 3). Values with different letters are significantly different at *p* < 0.05, as measured by a one-way ANOVA followed by a Duncan post hoc test.

**Figure 5 biomedicines-12-01980-f005:**
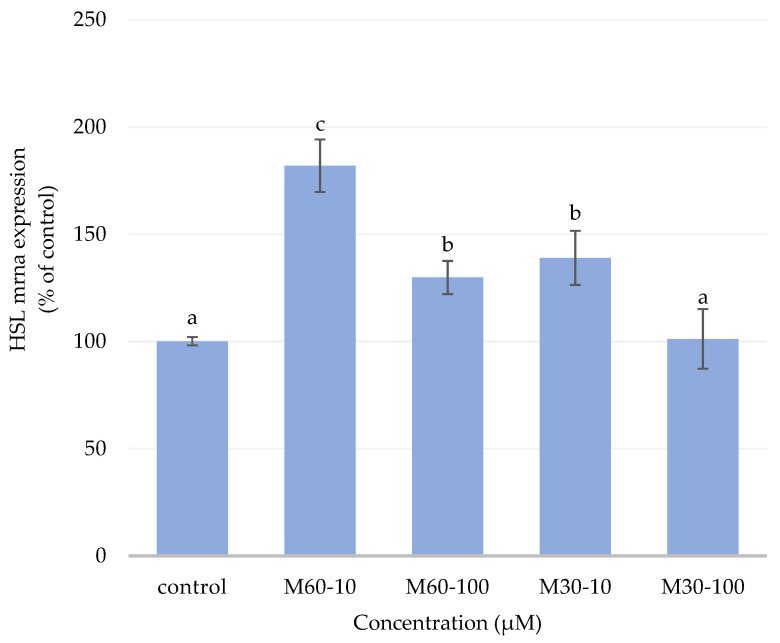
Semi-quantitative RT-PCR analysis for HSL mRNA levels in 3T3-L1 adipocytes after adding different ratios and concentrations of mixed oil for 72 h. The graph represents the mRNA levels for HSL relative to the control gene, β-actin. Values are expressed as the mean ± SEM (*n* = 3). Values with different letters are significantly different at *p* < 0.05, as measured by a one-way ANOVA followed by a Duncan post hoc test.

**Figure 6 biomedicines-12-01980-f006:**
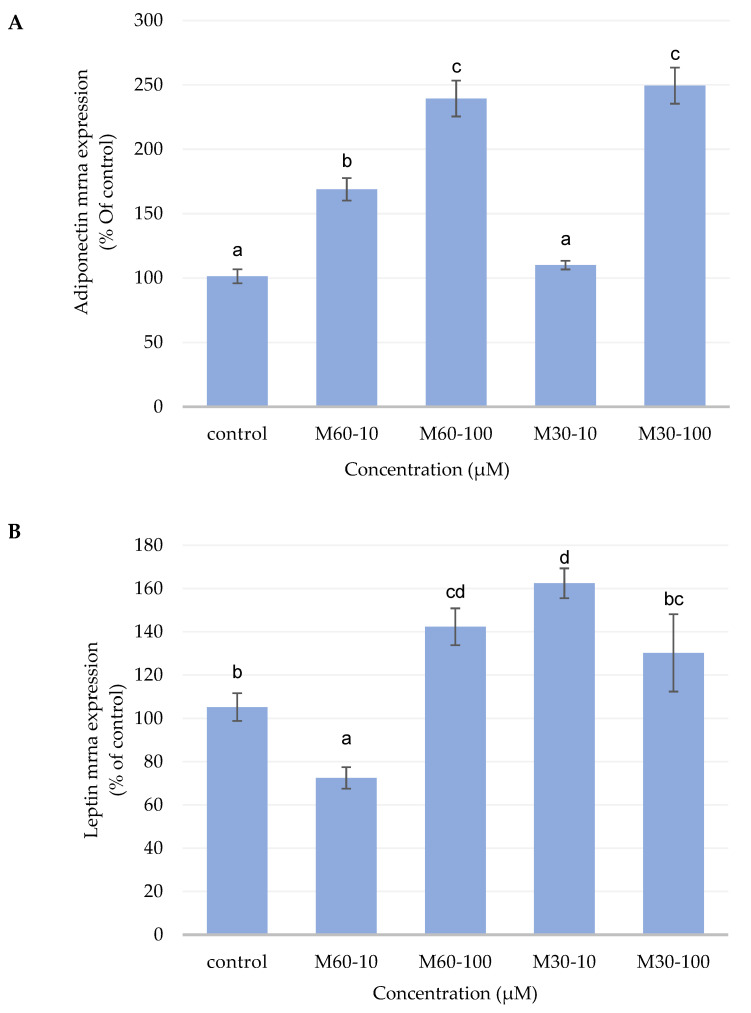
Semi-quantitative RT-PCR analysis for (**A**) leptin and (**B**) adiponectin mRNA levels in 3T3-L1 adipocytes after adding different ratios and concentrations of mixed oil for 72 h. The graph represents the mRNA levels for leptin and adiponectin relative to the control gene, β-actin. Values are expressed as the mean ± SEM (*n* = 3). Values with different letters are significantly different at *p* < 0.05, as measured by a one-way ANOVA followed by a Duncan post hoc test.

**Figure 7 biomedicines-12-01980-f007:**
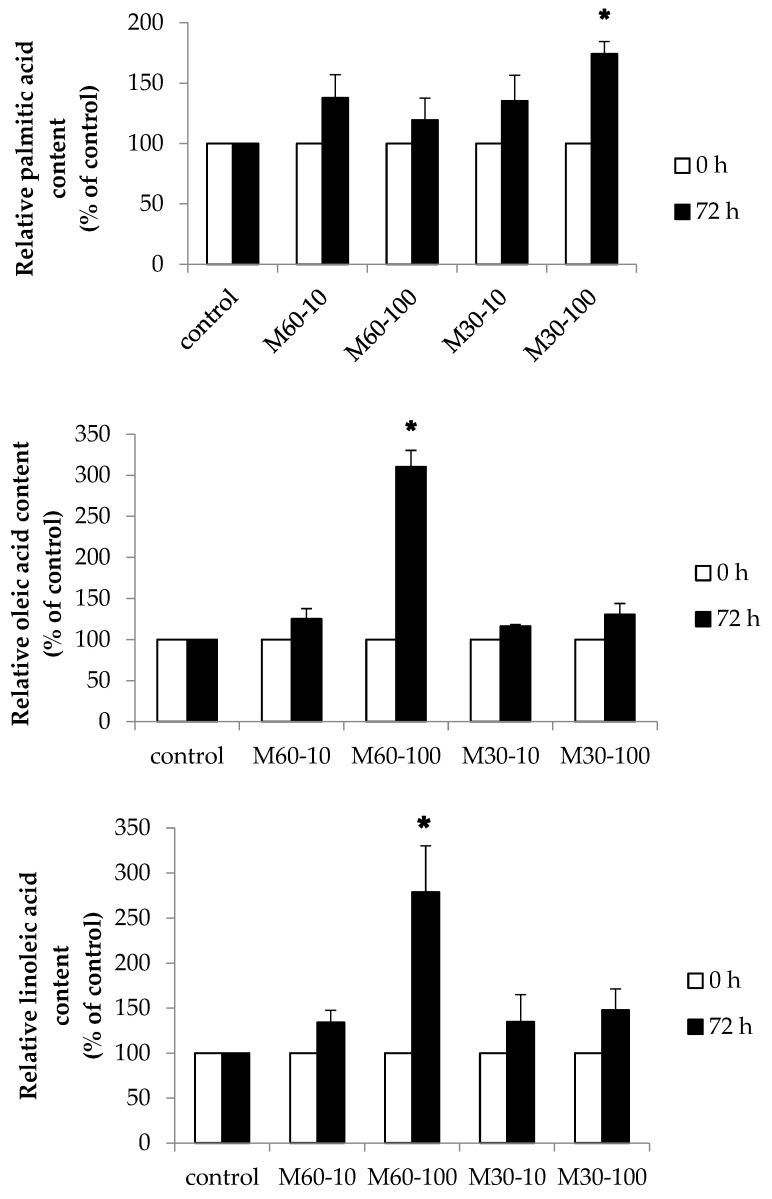
Relative fatty acid content in 3T3-L1 adipocytes after treatment with different ratios and concentrations of mixed oil for 72 h. Values are expressed as the mean ± SEM (*n* = 3). Values with * are significantly different at *p* < 0.05, as measured by a *t*-test.

**Figure 8 biomedicines-12-01980-f008:**
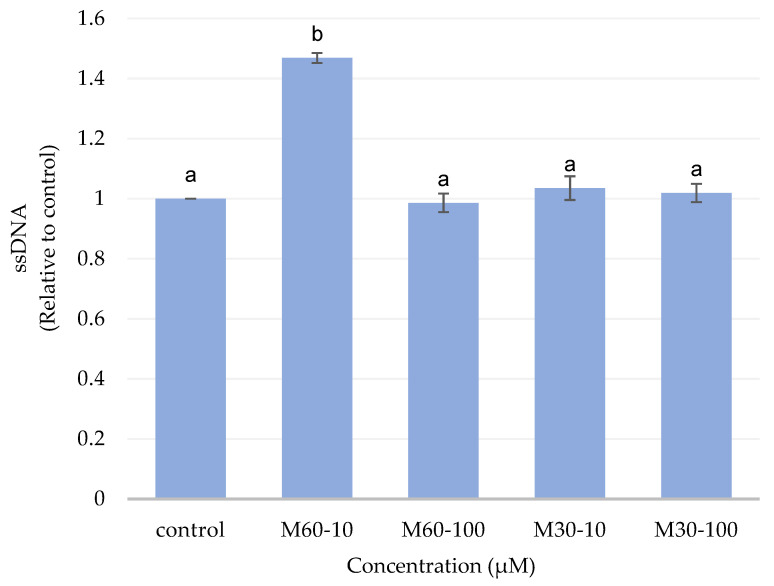
Effects of adding different ratios and concentrations of mixed fatty acids for 72 h on apoptosis in 3T3-L1 adipocytes. Values are expressed as the mean ± SEM (*n* = 3). Values with different letters are significantly different at *p* < 0.05, as measured by a one-way ANOVA followed by a Duncan post hoc test.

## Data Availability

The raw data supporting the conclusions of this article will be made available by the authors upon request.

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
