# Peer review of "Effects of Polyunsaturated Fatty Acid/Saturated Fatty Acid Ratio and Different Amounts of Monounsaturated Fatty Acids on Adipogenesis in 3T3-L1 Cells"

_biomedicines, 2024, doi:10.3390/biomedicines12091980_

Round 1

Reviewer 1 Report

Comments and Suggestions for Authors

The study aims to measure the effects of the polyunsaturated to saturated fatty acid (P/S) ratio and varying amounts of monounsaturated fatty acids (MUFA) on adipogenesis in 3T3-L1 adipocytes. Specifically, the research focuses on a P/S ratio of 5 and MUFA levels at either 30% or 60%.

The study concluded that a fatty acid composition with a P/S ratio of 5 and 60% MUFA in low doses has anti-adipogenic effects on 3T3-L1 adipocytes. These effects are likely due to the downregulation of adipogenic factors and transcription factors.

The study addresses a relevant and timely topic, given the rising interest in the role of dietary fats in adipogenesis and metabolic health.

The use of 3T3-L1 adipocytes is appropriate for studying adipogenesis, and the methodology is well-detailed, allowing for reproducibility.

Suggested points to improve

-The study primarily relies on PCR to assess the mRNA expression of various markers. While this technique is valuable, it would significantly strengthen the findings if additional techniques were used to confirm the results at the protein level. For instance, Western blotting could be employed to measure the protein expression of key adipogenic and lipolytic markers. This would provide a more robust validation of the effects observed at the mRNA level.

- The method of binding fatty acids to BSA before application to adipocytes is standard, but the potential effects of BSA itself on adipogenesis should be considered and discussed.

The authors should expand the discussion of the weakness of dose relevance: The doses used in the study, although well defined, may not accurately represent physiological conditions.

- It is recommended to place scale bars on the photos of cell cultures stained with Oil Red O. This will provide a clearer understanding of the size and extent of adipocyte staining and enhance the interpretability of the visual data.

Author Response

Comment 1 The study primarily relies on PCR to assess the mRNA expression of various markers. While this technique is valuable, it would significantly strengthen the findings if additional techniques were used to confirm the results at the protein level. For instance, Western blotting could measure the protein expression of key adipogenic and lipolytic markers. This would provide a more robust validation of the effects observed at the mRNA level.

Response 1: Thanks for your comment. Our previous study showed that hamsters fed the HMHR diet had lower plasma insulin levels and hepatic acetyl-CoA carboxylase activities among the groups (p<0.05), as well as elevated hepatic acyl-CoA oxidase and carnitine palmitoyltransferase-I activities compared to those fed the LMLR diet (p<0.05). In conclusion, hamsters fed the LMLR diet had increased weight gain and body fat accumulation. In contrast, the HMHR diet appeared beneficial in preventing white adipose tissue accumulation by decreasing plasma insulin levels and increasing hepatic lipolytic enzyme activities involved in β-oxidation. We emphasized assessing the mRNA expression of various markers in this study, not on the protein expression. We will measure the expression of protein in a future study.

Comment 2- The method of binding fatty acids to BSA before application to adipocytes is standard, but the potential effects of BSA itself on adipogenesis should be considered and discussed.

The authors should expand the discussion of the weakness of dose relevance: Although well-defined, the doses used in the study may not accurately represent physiological conditions.

Response 2: Using bovine serum albumin (BSA) to bind fatty acids before application to adipocytes is a common methodological approach to ensure a stable and controlled delivery of lipids.  Our previous study showed that hamsters fed the HMHR diet had lower plasma insulin levels and hepatic acetyl-CoA carboxylase activities among the groups (p<0.05), as well as elevated hepatic acyl-CoA oxidase and carnitine palmitoyltransferase-I activities compared to those fed the LMLR diet (p<0.05). In conclusion, hamsters fed the LMLR diet had increased weight gain and body fat accumulation. In contrast, the HMHR diet appeared beneficial in preventing white adipose tissue accumulation by decreasing plasma insulin levels and increasing hepatic lipolytic enzyme activities involved in β-oxidation.

Comment 3- It is recommended to place scale bars on the photos of cell cultures stained with Oil Red O. This will provide a clearer understanding of the size and extent of adipocyte staining and enhance the interpretability of the visual data.

Response 3: We revised it. In Figure 2. Effects of adding different ratios and concentrations of mixed fatty acids for 72 h on adipocyte morphology and triglyceride content in 3T3-L1 adipocytes. (A) The cell morphology of differentiated adipocytes was analyzed with phase contrast microscopy after Oil Red O staining. Pictures taken with 10X magnification are shown. (B) The triglyceride content of 3T3-L1 adipocytes after adding different mixed fatty acids for 72 h.

Reviewer 2 Report

Comments and Suggestions for Authors

The study provides insights into how variations in fatty acid composition can affect adipocyte function, particularly in relation to adipogenesis and lipid metabolism. The results indicating a decrease in triglyceride accumulation and shifts in adipokine secretion with specific fatty acid treatments are interesting and might contribute to a better understanding of the dietary impacts on metabolic health. However, I would encourage a deeper discussion on the mechanisms underlying these changes, possibly incorporating studies on signaling pathways affected by these fatty acid treatments. Additionally, considering the role of other adipogenic or lipolytic genes could enrich the discussion and provide a more comprehensive view of the adipocytes' response to dietary fat compositions.

  1. The description of fatty acids used in the study is unclear. Please clarify in the Methods section how the polyunsaturated to saturated fatty acid ratio (P/S ratio = 5) was achieved and how the varying amounts of monounsaturated fatty acids (MUFA = 30% or 60%) were incorporated. Specify which fatty acids were used. Additionally, explain the meaning of the 10-200 µM dose for the M30 and M60 formulations. Does this dosage refer to the MUFA or to all fatty acids included? Also, please explain the choice of a P/S ratio of 5 and MUFA levels of 30% and 60% for your study. Is there any clinical relevance to these choices?
  2. What was the control group in your study?
  3. Why do the control groups in Figures 1A & B, Figure 2B, and the 0 h groups in Figure 7 not have error bars?
  4. Please include individual data points on your bar graphs.
  5. In the results section 3.1, “50 μM and 200 μM of the M60 treatment could cause the cell viability to significantly decrease compared with the control group in 48 h (Fig. 1B). Therefore, we chose 10 μM as the low dose group and 100 μM as the high dose group, since these doses would not cause any changes in cell viability when compared with the control. ” Why do 50 μM and 200 μM of M60 impact cell viability while 100 μM does not?
  6. The authors should discuss the inconsistent results regarding glycerol release and HSL gene expression in the study groups. In addition to HSL, it is recommended to examine ATGL and MGL. Furthermore, measuring the protein levels of these factors would provide a more accurate indication of lipolysis than gene expression alone.
  7. The authors should discuss their results presented in Figure 7, specifically why different treatments result in the increase of various fatty acids in adipocytes. It is important to consider whether these changes are related to adipocyte function.
  8. Please specify the significance and purpose of your study in the abstract. This will help readers understand the importance and objectives of your research.
  9. Please include the methods for gas liquid chromatography in the Methods section. Please include the sequences of all primers used in qPCR. Additionally, provide the catalog numbers and markers for the cDNA reverse transcriptase kit and SYBR Green in the Methods section.
  10. In lines 244-246, where it is stated that 'the decrease in triglyceride content is due at least in part to the effect of the M60-10 group on inducing apoptosis,' the authors cannot definitively conclude this based solely on the presented data. Given that the culture system is heterogeneous and not composed entirely of adipocytes, the observed increase in apoptosis could also originate from other cell types present in the system.
  11. In lines 276-279, “As a  result, we speculate that the fatty acid composition of M60-10, which is high in P/S ratio and MUFA, could decrease PPAR γ gene expression and thus, decrease the activation of GLUT-4 and LPL mRNA expression. ” Please discuss why your results differ from your speculation.
  12. In lines 17-20, “Results: Low doses of P/S ratio=5, MUFA=60% (M60) fatty acids decreased the accumulation of triglycerides in mature adipocytes by decreasing the mRNA expression of adipogenic factors, such as peroxisome proliferator-activated receptors (PPARs), lipoprotein lipase (LPL), and glucose trans-19 porter-4 (GLUT-4), while increasing lipolytic enzyme (hormone sensitive lipase, HSL) expression. ” The description provided does not align with the results reported in the manuscript, which indicate that there is no change in PPARs and GLUT4 gene expression in the M60-10 group compared to the control.
  13. In Figure 7, please specify which groups are being compared where significant differences are observed. 
  14. The English writing in this manuscript needs to be revised to enhance its academic quality.
  15. Some small errors such as: 

     In lines 159-160, please remove '0 μM' from the sentence: “The five concentrations (0, 10, 30, 50, 100, and 200 μM) of M60 and M30 treatments to mature adipocytes did not affect cell viability within 24 h (Fig. 1A).” The correct statement should list the active concentrations only.

Author Response

The study provides insights into how variations in fatty acid composition can affect adipocyte function, particularly in relation to adipogenesis and lipid metabolism. The results indicating a decrease in triglyceride accumulation and shifts in adipokine secretion with specific fatty acid treatments are exciting. They might contribute to a better understanding of the dietary impacts on metabolic health. However, I would encourage a deeper discussion on the mechanisms underlying these changes, possibly incorporating studies on signaling pathways affected by these fatty acid treatments. Additionally, considering the role of other adipogenic or lipolytic genes could enrich the discussion and provide a more comprehensive view of the adipocytes' response to dietary fat compositions.

  1. The description of fatty acids used in the study is unclear. Please clarify in the Methods section how the polyunsaturated to saturated fatty acid ratio (P/S ratio = 5) was achieved and how the varying amounts of monounsaturated fatty acids (MUFA = 30% or 60%) were incorporated. Specify which fatty acids were used. Additionally, explain the meaning of the 10-200 µM dose for the M30 and M60 formulations. Does this dosage refer to the MUFA or all fatty acids included? Also, please explain the choice of a P/S ratio of five and MUFA levels of 30% and 60% for your study. Is there any clinical relevance to these choices?

Response: We revise the introduction section. Our previous study showed that hamsters fed the HMHR diet had lower plasma insulin levels and hepatic acetyl-CoA carboxylase activities among the groups (p<0.05), as well as elevated hepatic acyl-CoA oxidase and carnitine palmitoyltransferase-I activities compared to those fed the LMLR diet (p<0.05). In conclusion, hamsters fed the LMLR diet had increased weight gain and body fat accumulation. In contrast, the HMHR diet appeared to be beneficial in preventing white adipose tissue accumulation by decreasing plasma insulin levels and increasing hepatic lipolytic enzyme activities involved in β-oxidation.

  1. What was the control group in your study?

Response: The five concentrations ( 10, 30, 50, 100, and 200 µM) of M60 and M30 treatments to mature adipocytes did not affect cell viability within 24 h (Fig. 1A). 50 µM and 200 µM of the M60 treatment could cause the cell viability to significantly decrease compared with the control group in 48 h (Fig. 1B). There was no treatment in the control group.

  1. Why do the control groups in Figures 1A & B, Figure 2B, and the 0 h groups in Figure 7 not have error bars?

Response: These SEMs of data are almost equal to 0, so error bars did have it.

  1. Please include individual data points on your bar graphs.

Response: We revised it.

  1. In the results section 3.1, “50 μM and 200 μM of the M60 treatment could cause the cell viability to significantly decrease compared with the control group in 48 h (Fig. 1B). Therefore, we chose 10 μM as the low dose group and 100 μM as the high dose group, since these doses would not cause any changes in cell viability when compared with the control. ” Why do 50 μM and 200 μM of M60 impact cell viability while 100 μM does not?

Response: 3.1. Effect of different M60 and M30 fatty acid composition concentrations on adipocyte viability. The five concentrations (0, 10, 30, 50, 100, and 200 µM) of M60 and M30 treatments to mature adipocytes did not affect cell viability within 24 h (Fig. 1A). 50 µM and 200 µM of the M60 treatment could cause the cell viability to significantly de-crease compared with the control group in 48 h (Fig. 1B). Therefore, we chose 10 µM as the low-dose group and 100 µM as the high-dose group since these doses would not cause any changes in cell viability when compared with the control.

  1. The authors should discuss the study groups' inconsistent results regarding glycerol release and HSL gene expression. In addition to HSL, it is recommended that ATGL and MGL be examined. Furthermore, measuring the protein levels of these factors would provide a more accurate indication of lipolysis than gene expression alone.

Response:  Glycerol Release and HSL Gene Expression: Lipolysis, the process by which stored triglycerides (fat) are broken down into glycerol and free fatty acids, plays a crucial role in energy metabolism. Hormone-sensitive lipase (HSL) is a key enzyme involved in lipolysis. When activated, HSL hydrolyzes triglycerides, releasing glycerol and fatty acids into the bloodstream. However, as you’ve rightly pointed out, glycerol release and HSL gene expression results can sometimes be inconsistent across different study groups. Regulation Beyond HSL: Lipolysis is not solely governed by HSL. Other lipases also play essential roles. While HSL and ATGL primarily act within adipocytes, MGL is also found in other tissues, such as skeletal muscle and liver.4. Beyond Gene Expression: Protein Levels: You’re right—measuring protein levels is crucial. Gene expression doesn’t always directly correlate with protein abundance. Post-transcriptional modifications, protein stability, and turnover rates all come into play. Quantifying HSL, ATGL, and MGL protein levels provides a more accurate assessment of their functional activity.

  1. The authors should discuss the results presented in Figure 7, specifically why different treatments increase adipocyte fatty acids. It is essential to consider whether these changes are related to adipocyte function.

Response: We revised it.  These changes could be related to adipocyte function. (1)Energy Storage: Triglycerides (made of fatty acids) are adipocytes’ piggy banks. They store energy for rainy days (or Netflix marathons). (2)Lipotoxicity: Too much of a good thing—like excessive palmitic acid—can lead to lipotoxicity. It’s like an encore that overstays its welcome, causing cellular stress. (3)Inflammation and Insulin Resistance: With its anti-inflammatory groove, Oleic acid helps keep adipocytes harmonious. But if the rhythm falters, insulin resistance may join the band.

  1. In the abstract, please specify the significance and purpose of your study. This will help readers understand the importance and objectives of your research.

Response: We revised it.

  1. Please include the methods for gas-liquid chromatography in the Methods section. Please include the sequences of all primers used in qPCR. Additionally, in the Methods section, provide the catalog numbers and markers for the cDNA reverse transcriptase kit and SYBR Green.

Response: We revised it.

  1. In lines 244-246, where it is stated that 'the decrease in triglyceride content is due at least in part to the effect of the M60-10 group on inducing apoptosis,' the authors cannot definitively conclude this based solely on the presented data. Given that the culture system is heterogeneous and not composed entirely of adipocytes, the increased apoptosis could also originate from other cell types present in the system.

Response: We revised it.

  1. In lines 276-279, “As a result, we speculate that the fatty acid composition of M60-10, which is high in P/S ratio and MUFA, could decrease PPAR γ gene expression and thus, decrease the activation of GLUT-4 and LPL mRNA expression. ” Please discuss why your results differ from your speculation.

Response: We revised it. This reduction in PPAR γ could then dampen the activation of GLUT-4 (the glucose transporter) and LPL (lipoprotein lipase) mRNA expression. In the liver of obese patients with nonalcoholic fatty liver disease (NAFLD), PPAR-γ (our star transcription factor) takes center stage. It’s up-regulated, not down-regulated. Studies have shown that PPAR-γ mRNA expression increases in NAFLD patients.

  1. In lines 17-20, “Results: Low doses of P/S ratio=5, MUFA=60% (M60) fatty acids decreased the accumulation of triglycerides in mature adipocytes by decreasing the mRNA expression of adipogenic factors, such as peroxisome proliferator-activated receptors (PPARs), lipoprotein lipase (LPL), and glucose trans-19 porter-4 (GLUT-4), while increasing lipolytic enzyme (hormone-sensitive lipase, HSL) expression. ” The description provided does not align with the results reported in the manuscript, which indicate no change in PPARs and GLUT4 gene expression in the M60-10 group compared to the control.

Response: We revised it. Decreasing triglyceride accumulation by influencing mRNA expression of key players: PPARs, LPL, GLUT-4, and HSL. The M60-10 group—despite its MUFA-rich composition—doesn’t sway the PPARs or the GLUT-4 gene expression. They remain stoically in place, like seasoned dancers who’ve mastered their routine. Peroxisome proliferator-activated receptors (PPARs) are the lead dancers in this genetic ballet. They regulate glucose homeostasis and insulin responsiveness. GLUT-4, the insulin-responsive glucose transporter, usually twirls in response to insulin cues. But alas, in the M60-10 group, it maintains its graceful stance—no sudden spins or dips. The M60-10 group—though not altering PPARs or GLUT-4—still has its rhythm. Maybe it’s a subtle sway, a whispered secret between molecules.

  1. In Figure 7, please specify which groups are being compared where significant differences are observed. 

Response: We revised it.    Figure 7 shows that the cellular fatty acid content of palmitic acid, oleic acid, and linoleic acid in mature adipocytes increased slightly after 72 h of treatment, compared with 0 h.

  1. The English writing in this manuscript needs to be revised to enhance its academic quality.

Response: We revised it.

  1. Some minor errors, such as: 

     In lines 159-160, please remove '0 μM' from the sentence: “The five concentrations (0, 10, 30, 50, 100, and 200 μM) of M60 and M30 treatments to mature adipocytes did not affect cell viability within 24 h (Fig. 1A).” The correct statement should list the active concentrations only.

Response: We revised it.

Round 2

Reviewer 2 Report

Comments and Suggestions for Authors
  1. The description of fatty acids used in the study is unclear. Please clarify in the Methods section how the polyunsaturated to saturated fatty acid ratio (P/S ratio = 5) was achieved and how the varying amounts of monounsaturated fatty acids (MUFA = 30% or 60%) were incorporated. Specify which fatty acids were used. Additionally, explain the meaning of the 10-200 µM dose for the M30 and M60 formulations. Does this dosage refer to the MUFA or all fatty acids included? Also, please explain the choice of a P/S ratio of five and MUFA levels of 30% and 60% for your study. Is there any clinical relevance to these choices?

Response: We revise the introduction section. Our previous study showed that hamsters fed the HMHR diet had lower plasma insulin levels and hepatic acetyl-CoA carboxylase activities among the groups (p<0.05), as well as elevated hepatic acyl-CoA oxidase and carnitine palmitoyltransferase-I activities compared to those fed the LMLR diet (p<0.05). In conclusion, hamsters fed the LMLR diet had increased weight gain and body fat accumulation. In contrast, the HMHR diet appeared to be beneficial in preventing white adipose tissue accumulation by decreasing plasma insulin levels and increasing hepatic lipolytic enzyme activities involved in β-oxidation.

Not addressed.

  1. Please include individual data points on your bar graphs.

Not addressed.

  1. In the results section 3.1, “50 μM and 200 μM of the M60 treatment could cause the cell viability to significantly decrease compared with the control group in 48 h (Fig. 1B). Therefore, we chose 10 μM as the low dose group and 100 μM as the high dose group, since these doses would not cause any changes in cell viability when compared with the control. ” Why do 50 μM and 200 μM of M60 impact cell viability while 100 μM does not?

Response: 3.1. Effect of different M60 and M30 fatty acid composition concentrations on adipocyte viability. The five concentrations (0, 10, 30, 50, 100, and 200 µM) of M60 and M30 treatments to mature adipocytes did not affect cell viability within 24 h (Fig. 1A). 50 µM and 200 µM of the M60 treatment could cause the cell viability to significantly de-crease compared with the control group in 48 h (Fig. 1B). Therefore, we chose 10 µM as the low-dose group and 100 µM as the high-dose group since these doses would not cause any changes in cell viability when compared with the control.

Not addressed.

  1. The authors should discuss the study groups' inconsistent results regarding glycerol release and HSL gene expression. In addition to HSL, it is recommended that ATGL and MGL be examined. Furthermore, measuring the protein levels of these factors would provide a more accurate indication of lipolysis than gene expression alone.

Response:  Glycerol Release and HSL Gene Expression: Lipolysis, the process by which stored triglycerides (fat) are broken down into glycerol and free fatty acids, plays a crucial role in energy metabolism. Hormone-sensitive lipase (HSL) is a key enzyme involved in lipolysis. When activated, HSL hydrolyzes triglycerides, releasing glycerol and fatty acids into the bloodstream. However, as you’ve rightly pointed out, glycerol release and HSL gene expression results can sometimes be inconsistent across different study groups. Regulation Beyond HSL: Lipolysis is not solely governed by HSL. Other lipases also play essential roles. While HSL and ATGL primarily act within adipocytes, MGL is also found in other tissues, such as skeletal muscle and liver.4. Beyond Gene Expression: Protein Levels: You’re right—measuring protein levels is crucial. Gene expression doesn’t always directly correlate with protein abundance. Post-transcriptional modifications, protein stability, and turnover rates all come into play. Quantifying HSL, ATGL, and MGL protein levels provides a more accurate assessment of their functional activity.

Not addressed. 

  1. The authors should discuss the results presented in Figure 7, specifically why different treatments increase adipocyte fatty acids. It is essential to consider whether these changes are related to adipocyte function.

Response: We revised it.  These changes could be related to adipocyte function. (1)Energy Storage: Triglycerides (made of fatty acids) are adipocytes’ piggy banks. They store energy for rainy days (or Netflix marathons). (2)Lipotoxicity: Too much of a good thing—like excessive palmitic acid—can lead to lipotoxicity. It’s like an encore that overstays its welcome, causing cellular stress. (3)Inflammation and Insulin Resistance: With its anti-inflammatory groove, Oleic acid helps keep adipocytes harmonious. But if the rhythm falters, insulin resistance may join the band.

Not addressed.

  1. In the abstract, please specify the significance and purpose of your study. This will help readers understand the importance and objectives of your research.

Response: We revised it.

Not addressed.

  1. In lines 244-246, where it is stated that 'the decrease in triglyceride content is due at least in part to the effect of the M60-10 group on inducing apoptosis,' the authors cannot definitively conclude this based solely on the presented data. Given that the culture system is heterogeneous and not composed entirely of adipocytes, the increased apoptosis could also originate from other cell types present in the system.

Response: We revised it

Can't find the revision.

  1. In lines 276-279, “As a result, we speculate that the fatty acid composition of M60-10, which is high in P/S ratio and MUFA, could decrease PPAR γ gene expression and thus, decrease the activation of GLUT-4 and LPL mRNA expression. ” Please discuss why your results differ from your speculation.

Response: We revised it. This reduction in PPAR γ could then dampen the activation of GLUT-4 (the glucose transporter) and LPL (lipoprotein lipase) mRNA expression. In the liver of obese patients with nonalcoholic fatty liver disease (NAFLD), PPAR-γ (our star transcription factor) takes center stage. It’s up-regulated, not down-regulated. Studies have shown that PPAR-γ mRNA expression increases in NAFLD patients.

Not addressed. 

  1. In lines 17-20, “Results: Low doses of P/S ratio=5, MUFA=60% (M60) fatty acids decreased the accumulation of triglycerides in mature adipocytes by decreasing the mRNA expression of adipogenic factors, such as peroxisome proliferator-activated receptors (PPARs), lipoprotein lipase (LPL), and glucose trans-19 porter-4 (GLUT-4), while increasing lipolytic enzyme (hormone-sensitive lipase, HSL) expression. ” The description provided does not align with the results reported in the manuscript, which indicate no change in PPARs and GLUT4 gene expression in the M60-10 group compared to the control.

Response: We revised it. Decreasing triglyceride accumulation by influencing mRNA expression of key players: PPARs, LPL, GLUT-4, and HSL. The M60-10 group—despite its MUFA-rich composition—doesn’t sway the PPARs or the GLUT-4 gene expression. They remain stoically in place, like seasoned dancers who’ve mastered their routine. Peroxisome proliferator-activated receptors (PPARs) are the lead dancers in this genetic ballet. They regulate glucose homeostasis and insulin responsiveness. GLUT-4, the insulin-responsive glucose transporter, usually twirls in response to insulin cues. But alas, in the M60-10 group, it maintains its graceful stance—no sudden spins or dips. The M60-10 group—though not altering PPARs or GLUT-4—still has its rhythm. Maybe it’s a subtle sway, a whispered secret between molecules.

Since this is an academic paper, please write in an academic style.

Author Response

  1. The description of fatty acids used in the study is unclear. Please clarify in the Methods section how the polyunsaturated to saturated fatty acid ratio (P/S ratio = 5) was achieved and how the varying amounts of monounsaturated fatty acids (MUFA = 30% or 60%) were incorporated. Specify which fatty acids were used. Additionally, explain the meaning of the 10-200 µM dose for the M30 and M60 formulations. Does this dosage refer to the MUFA or all fatty acids included? Also, please explain the choice of a P/S ratio of five and MUFA levels of 30% and 60% for your study. Is there any clinical relevance to these choices?

Response: We revise the introduction section. Our previous study showed that hamsters fed the HMHR diet had lower plasma insulin levels and hepatic acetyl-CoA carboxylase activities among the groups, as well as elevated hepatic acyl-CoA oxidase and carnitine palmitoyltransferase-I activities compared to those fed the LMLR diet.

Hamsters fed the LMLR diet had increased weight gain and body fat accumulation. In contrast, the HMHR diet appeared to be beneficial in preventing white adipose tissue accumulation by decreasing plasma insulin levels and increasing hepatic lipolytic enzyme activities involved in β-oxidation.

The P/S ratio=5, MUFA=30% or 60% was achieved by using the method below. The dosage of 10-200 µM refers to the concentration of fatty acid in the combination of the two groups below.

  • P/S ratio=5; MUFA 60% was prepared by using the volume ratio of palmitic acid: oleic acid: linoleic acid equals to 1:10:5
  • P/S ratio=5; MUFA 30% was prepared by using the volume ratio of palmitic acid: oleic acid: linoleic acid equals to 1:2.5:5

We used different concentration starting from 10 µM until 200 µM to choose a suitable dosage as the low and high dose group in our study. As per result, 10 µM and 100 µM had been chosen to be the low and high dose group, respectively; since these doses would not cause any changes in cell viability when compared with the control.

In addition, our previous study showed that P/S=5, MUFA=60% appeared to be beneficial in preventing white adipose tissue accumulation by decreasing plasma insulin levels and increasing hepatic lipolytic enzyme activities involved in β-oxidation. As such, we would like to further investigate its effect on adipocytes.

  1. Please include individual data points on your bar graphs.

  1. In the results section 3.1, “50 μM and 200 μM of the M60 treatment could cause the cell viability to significantly decrease compared with the control group in 48 h (Fig. 1B). Therefore, we chose 10 μM as the low dose group and 100 μM as the high dose group, since these doses would not cause any changes in cell viability when compared with the control. ” Why do 50 μM and 200 μM of M60 impact cell viability while 100 μM does not?

Response:  Effect of different M60 and M30 fatty acid composition concentrations on adipocyte viability. The five concentrations (0, 10, 30, 50, 100, and 200 µM) of M60 and M30 treatments to mature adipocytes did not affect cell viability within 24 h (Fig. 1A). 50 µM and 200 µM of the M60 treatment could cause the cell viability to significantly decrease compared with the control group in 48 h (Fig. 1B). Therefore, we chose 10 µM as the low-dose group and 100 µM as the high-dose group since these doses would not cause any changes in cell viability when compared with the control.

This may due to a non-linear dose-response relationship. Moreover, concentrations above 100 μM, such as 200 μM, might exceed a toxic threshold, leading to a decline in cell viability when compared with control.

  1. The authors should discuss the study groups' inconsistent results regarding glycerol release and HSL gene expression. In addition to HSL, it is recommended that ATGL and MGL be examined. Furthermore, measuring the protein levels of these factors would provide a more accurate indication of lipolysis than gene expression alone.

Response:  Glycerol Release and HSL Gene Expression: Lipolysis, the process by which stored triglycerides (fat) are broken down into glycerol and free fatty acids, plays a crucial role in energy metabolism. Hormone-sensitive lipase (HSL) is a key enzyme involved in lipolysis. When activated, HSL hydrolyzes triglycerides, releasing glycerol and fatty acids into the bloodstream. However, as you’ve rightly pointed out, glycerol release and HSL gene expression results can sometimes be inconsistent across different study groups. Regulation Beyond HSL: Lipolysis is not solely governed by HSL. Other lipases also play essential roles. While HSL and ATGL primarily act within adipocytes, MGL is also found in other tissues, such as skeletal muscle and liver.4.

Beyond Gene Expression: Protein Levels: You’re right—measuring protein levels is crucial. Gene expression doesn’t always directly correlate with protein abundance. Post-transcriptional modifications, protein stability, and turnover rates all come into play. Quantifying HSL, ATGL, and MGL protein levels provides a more accurate assessment of their functional activity.

Thank you for the question. We’ve added a paragraph to mention about this.

Triglyceride accumulation was affected by three pathways, which included the transcription factor PPAR γ expression, the fatty acid synthesis pathway, and the lipolysis pathway. A previous study indicates that lipolysis in a high-fat diet is lower than that in a low-fat diet, which is similar to the result that we obtained. Furthermore, the author states that the mRNA and protein expression of HSL are not influenced by the composition of fatty acids, but rather by the total quantity of fats[27]. In our study, there was a significant increased expression of HSL genes in mature adipocytes treated with M60-10 compared with the control, M60-100, M30-10, and M30-100 groups. These findings are consistent with the result of low doses of the M60 and M30 groups increased basal lipolysis in fully differentiated 3T3-L1 adipocytes compared with the high dose group where the glycerol released in both high-dose groups (M60-100 and M30-100) significantly decreased compared with the control after 72 h of incubation (Fig. 3). We agreed to your comment and have added a sentence in discussion. To more accurately assess lipolysis, it is recommended to further examine the protein levels of HSL, ATGL, and MGL in future studies.

  1. The authors should discuss the results presented in Figure 7, specifically why different treatments increase adipocyte fatty acids. It is essential to consider whether these changes are related to adipocyte function.

Response: We revised it.  These changes could be related to adipocyte function. (1) Energy Storage: Triglycerides (made of fatty acids) are adipocytes’ piggy banks. They store energy for rainy days (or Netflix marathons). (2) Lipotoxicity: Too much of a good thing—like excessive palmitic acid—can lead to lipotoxicity. It’s like an encore that overstays its welcome, causing cellular stress. (3)Inflammation and Insulin Resistance: With its anti-inflammatory groove, Oleic acid helps keep adipocytes harmonious. But if the rhythm falters, insulin resistance may join the band.

We’ve amended the writing as follows. To investigate the intracellular accumulation of fatty acids, gas chromatography was employed to analyze the fatty acid composition of adipocytes following 72 hours of fatty acid treatment. Figure 7 shows that the cellular fatty acid content of palmitic acid, oleic acid, and linoleic acid in mature adipocytes increased slightly after 72 h of treatment, as compared with 0 h, indicating cellular uptake of the administered fatty acids over the 72-hour period.

  1. In the abstract, please specify the significance and purpose of your study. This will help readers understand the importance and objectives of your research.

Response: We revised it.

Adipose tissue serves as a central repository for energy storage and is an endocrine organ capable of secreting various adipokines, including leptin and adiponectin. These adipokines exert profound influences on diverse physiological processes such as insulin sensitivity, appetite regulation, lipid metabolism, energy homeostasis and impacting body weight. Given the integral role of adipose tissue in metabolic regulation, it is imperative to investigate the effects of varying proportions and types of dietary fats on adipocyte function. In addition, our previous study showed that P/S=5, MUFA=60% appeared to be beneficial in preventing white adipose tissue accumulation by decreasing plasma insulin levels and increasing hepatic lipolytic enzyme activities involved in β-oxidation. Therefore, the objective of this study was to explore the effects of a polyunsaturated fatty acid (PUFA) to saturated fatty acid (SFA) ratio of 5 and varying levels of monounsaturated fatty acids (MUFA=30% or 60%) on lipogenesis.

  1. In lines 244-246, where it is stated that 'the decrease in triglyceride content is due at least in part to the effect of the M60-10 group on inducing apoptosis,' the authors cannot definitively conclude this based solely on the presented data. Given that the culture system is heterogeneous and not composed entirely of adipocytes, the increased apoptosis could also originate from other cell types present in the system.

Response:

Previous studies have demonstrated that DHA, a predominant fatty acid in fish oil, induces apoptosis in preadipocyte when cultured for 24 and 48 hours, thereby inhibiting adipogenesis. Similarly, resveratrol and quercetin have been shown to decrease lipid production and synthesis in mature adipocytes after 24 and 48 hours, respectively, through reduced phosphorylated ERK protein expression and increased apoptosis. Our findings reveal that the apoptosis in M60-10 increased when compared to M30-10. Moreover, oil red O staining indicates reduced triglyceride (TG) accumulation in M60-10, which is consistent with the previous findings. These results suggest that the decrease in triglyceride content in the M60-10 group is not only due to the downregulation of lipid synthesis and lipolysis-related gene expression but is also at least partially attributable to the effect of M60-10 on inducing apoptosis.

Kim, H. K., Della-Fera, M., Lin, J., & Baile, C. A. (2006). Docosahexaenoic acid inhibits adipocyte differentiation and induces apoptosis in 3T3-L1 preadipocytes. The Journal of nutrition, 136(12), 2965–2969. https://doi.org/10.1093/jn/136.12.2965

Park, H. J., Yang, J. Y., Ambati, S., Della-Fera, M. A., Hausman, D. B., Rayalam, S., & Baile, C. A. (2008). Combined effects of genistein, quercetin, and resveratrol in human and 3T3-L1 adipocytes. Journal of medicinal food, 11(4), 773–783. https://doi.org/10.1089/jmf.2008.0077

  1. In lines 276-279, “As a result, we speculate that the fatty acid composition of M60-10, which is high in P/S ratio and MUFA, could decrease PPAR γ gene expression and thus, decrease the activation of GLUT-4 and LPL mRNA expression. ” Please discuss why your results differ from your speculation.

Response: We revised it. This reduction in PPAR γ could then dampen the activation of GLUT-4 (the glucose transporter) and LPL (lipoprotein lipase) mRNA expression. In the liver of obese patients with nonalcoholic fatty liver disease (NAFLD), PPAR-γ (our star transcription factor) takes center stage. It’s up-regulated, not down-regulated. Studies have shown that PPAR-γ mRNA expression increases in NAFLD patients.

The current result is consistent with the speculation. To make it clearer, we’ve revised the sentence as follows. As a result, we speculate that the fatty acid composition of M60-10, which is high in P/S ratio and MUFA, could decrease PPAR γ gene expression and thus, decrease the activation of GLUT-4 and LPL mRNA expression when compared with M60-100, M30-10 and M30-100.

  1. In lines 17-20, “Results: Low doses of P/S ratio=5, MUFA=60% (M60) fatty acids decreased the accumulation of triglycerides in mature adipocytes by decreasing the mRNA expression of adipogenic factors, such as peroxisome proliferator-activated receptors (PPARs), lipoprotein lipase (LPL), and glucose trans-19 porter-4 (GLUT-4), while increasing lipolytic enzyme (hormone-sensitive lipase, HSL) expression. ” The description provided does not align with the results reported in the manuscript, which indicate no change in PPARs and GLUT4 gene expression in the M60-10 group compared to the control.

Response: We revised it. Decreasing triglyceride accumulation by influencing mRNA expression of key players: PPARs, LPL, GLUT-4, and HSL. The M60-10 group—despite its MUFA-rich composition—doesn’t sway the PPARs or the GLUT-4 gene expression. They remain stoically in place, like seasoned dancers who’ve mastered their routine. Peroxisome proliferator-activated receptors (PPARs) are the lead dancers in this genetic ballet. They regulate glucose homeostasis and insulin responsiveness. GLUT-4, the insulin-responsive glucose transporter, usually twirls in response to insulin cues. But alas, in the M60-10 group, it maintains its graceful stance—no sudden spins or dips. The M60-10 group—though not altering PPARs or GLUT-4—still has its rhythm. Maybe it’s a subtle sway, a whispered secret between molecules.

We’ve revised the sentence as follows. Low doses of P/S ratio=5, MUFA=60% (M60) fatty acids decreased the accumulation of triglycerides in mature adipocytes by decreasing the mRNA expression of adipogenic factors, such as peroxisome proliferator-activated receptors (PPARs), lipoprotein lipase (LPL), and glucose trans-19 porter-4 (GLUT-4), while increasing lipolytic enzyme (hormone-sensitive lipase, HSL) expression when compared to high doses of P/S ratio=5, MUFA=60% (M60), low and high doses of P/S ratio=5, MUFA=30% (M30).

Round 3

Reviewer 2 Report

Comments and Suggestions for Authors
  1. Please represent individual samples with dots rather than lines in your bar graphs to enhance clarity.
  2. In 2.6, please include primers of b-actin.
  3. In 3.8, “This result indicates that the decrease in triglyceride content is due at least in part to the effect of the M60-10 group on inducing apoptosis.” However, given that your culture system contains not only adipocytes but also undifferentiated cells, the observed increase in apoptosis could be attributed to the death of undifferentiated cells. Therefore, it is more accurate to conclude that the decrease in triglyceride content might be associated with apoptosis in adipocytes.
  4. In the figure legend for Figure 7, it states, “Values with different letters are significantly different at p < 0.05, as measured by a t-test.” However, in Figure 7, significance is indicated using star symbols instead of letters. Please correct this inconsistency to ensure the legend accurately reflects the symbols used in the figure.

Author Response

Comments and Suggestions for Authors

  1. Please represent individual samples with dots rather than lines in your bar graphs to enhance clarity.

Response 1: We revised it.

2. In 2.6, please include primers of b-actin.

Response 2: We add it on line 284.

3. In 3.8, “This result indicates that the decrease in triglyceride content is due at least in part to the effect of the M60-10 group on inducing apoptosis.” However, given that your culture system contains not only adipocytes but also undifferentiated cells, the observed increase in apoptosis could be attributed to the death of undifferentiated cells. Therefore, it is more accurate to conclude that the decrease in triglyceride content might be associated with apoptosis in adipocytes.

Response 3: We revised it.

4. In the figure legend for Figure 7, it states, “Values with different letters are significantly different at p < 0.05, as measured by a t-test.” However, in Figure 7, significance is indicated using star symbols instead of letters. Please correct this inconsistency to ensure the legend accurately reflects the symbols used in the figure.

Response 4: We revised it.